# Intelligent Model Update Strategy for Sequential Recommendation

## ABSTRACT

Modern online platforms are increasingly employing recommendation systems to address information overload and improve user engagement. There is an evolving paradigm in this research field that recommendation network learning occurs both on the cloud and on edge edges with knowledge transfer in between (*i.e.*, edge-cloud collaboration). Recent works push this filed further by enabling edge-specific context-aware adaptivity, where model parameters are updated in real-time based on incoming on-edge data. However, we argue that frequent data exchanges between the cloud and edges often lead to inefficiency and waste of communication/computation resources, as considerable parameter updates might be redundant. To investigate this problem, we introduce **Intell**igent **E**dge-**C**loud Parame**t**er **Req**uest Model (**IntellectReq**[1]). IntellectReq is designed to operate on edge, evaluating the cost-benefit landscape of parameter requests with minimal computation and communication overhead. We formulate this as a novel learning task, aimed at the detection of out-of-distribution data, thereby fine-tuning adaptive communication strategies. Further, we employ statistical mapping techniques to convert real-time user behavior into a normal distribution, thereby employing multi-sample outputs to quantify the model's uncertainty and thus its generalization capabilities. Rigorous empirical validation on four widely-adopted benchmarks evaluates our approach, evidencing a marked improvement in the efficiency and generalizability of edge-cloud collaborative and dynamic recommendation systems.

## CCS CONCEPTS

• **Information systems** → **Mobile information processing systems**; **Personalization**; • **Human-centered computing** → **Mobile computing**.

## KEYWORDS

Edge-Cloud Collaboration, Mis-Recommendation Detection, Out-of-Distribution Detection, Sequential Recommendation

## 1 INTRODUCTION

With the rapid development of e-commerce and social media platforms, recommendation systems have become indispensable tools in people's daily life. They can be recognized as various forms depending on industries, like product suggestions on online e-commerce websites, *e.g.*, Amazon and Taobao) or playlist generators for video and music services (*e.g.*, YouTube, Netflix, and Spotify). Among them, one of the classical recommendation systems in the industry prefers to trains a universal model with static parameters on a powerful cloud conditioned on rich data collected from different edges, and then perform edge inference for all users, such as *e.g.*, DIN [20], SASRec [6], and GRU4Rec [5]. As the first model in Figure 1, this form of cloud static model lets users share a centralized model that

enables real-time inference for all edges, but fails to exploit the personalized recommendation pattern for each particular edge due to the data distribution shift between cloud and edge.

To alleviate this issue, existing solutions can be summarized into two lines: (i) *Edge-cloud Collaboration* [2] [17, 18]: To access personalization, the second model in Figure 1(a) enables on-edge learning given the centralized recommendation model. Such as distillation [12] and fine-tuning [2] methods can eliminate edge-cloud distribution shift based on extra training on the edges. However, the edge retraining incurs numerous calculations on the gradients to update the model parameters, which is undesirable when the edge applications typically have the real-time requirement constraint. (ii) *Real-time Dynamic Recommendation*: Most recently, to realize the personalization and real-time requirement, an advanced approach called dynamic parameters generation [7, 16] (third model in Figure 1(a)) entails *low calculation cost* on-edge learning for model personalization. Specifically, it maps the real-time user's click sequence to adaptive parameters through forward propagation of a trained hypernetwork [4]. The generated parameters can be deployed on the cloud model that measures the real-time data distribution for fast recommendation personalization. This recommendation learning paradigm can be regarded as the **D**evice-**C**loud **C**ollaborative and **D**ynamic **R**ecommendation system (**DC-CDR**), which enables the personalized recommendation pattern for different edges and efficiently characterizes the real-time data distribution based on frequent edge-cloud communication.

Despite promising, DC-CDR cannot be easily deployed in the real environment, due to the two key aspects summarized as follows:

- **High Request Frequency.** Once a new data sequence is clicked by the user on the edge, the DC-CDR model will update the model parameters through the edge-cloud communication. In industrial scenarios, it prohibitively results in a large number of edges requesting the cloud concurrently. The situation is further exacerbated when the networking environment is unstable, which limits the DC-CDR's efficiency under such communication and network constraints.

- **Low Communication Revenue.** In addition, the edge-cloud communication is unnecessary when the latest data corresponded to the current model's parameters and the real-time data obey the same data distribution, *i.e.*, the distribution shift occasions are not always consistent with the on-cloud parameters requests. Those unnecessary communications' between the cloud and edge would potentially cause the over-consumption of communication resources with low revenue, hindering the practicality of the DC-CDR.

To further access the communication problem in DC-CDR, we analyze the user's click class (can be regarded as domain) on the edges. We first collected the item embedding vectors in the user

---

[1]Our project is available on https://anonymous.4open.science/r/IntellectReq-0628/

[2]This paper considers edge-cloud collaboration from the perspective of on-edge learning.

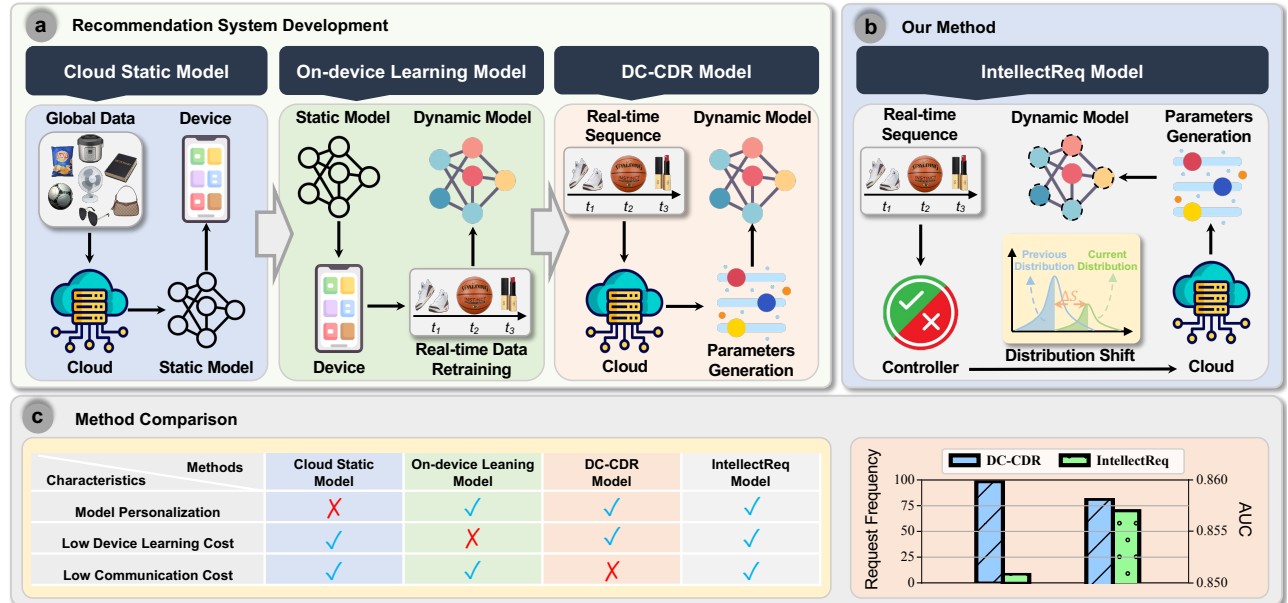

Figure 1: (a) describes the developing trend of recommendation systems that evolved from cloud static model to DC-CDR model. (b) overviews our proposed high-efficiency DC-CDR, *i.e.*, IntellectReq. (c) shows the comparison of characteristics of four recommendation systems and the communication cost of DC-CDR and our IntellectReq (`Communication Frequency 10% (IntellectReq) ≪ 100% (DC-CDR)`, AUC: 0.8562 (IntellectReq) ≈ 0.8581 (DC-CDR)).

click sequence from four public datasets and then classify them into 50 domains. As shown in Figure 2, only 10∼15 domains are included in the long user sequence in most cases, which means that users often repeatedly click on items belonging to some specific domains. However, DC-CRS cannot detect that the data distribution shift on the edge, which leads to a highly frequent request of dynamic parameters, along with the excessive communication consumption.

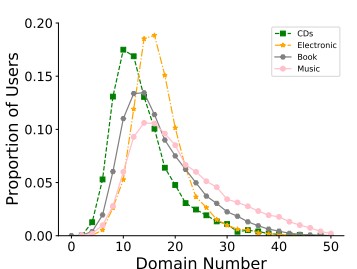

Figure 2: Domain numbers of users.

Based on the aforementioned insights, a valuable optimization goal is to reduce unnecessary communications, thereby yielding a high-efficiency DC-CDR system. To access this goal, as described in 1(b), we designed an **I**ntelligent **DE**vice-Cloud P**A**rameter Request Mode**L** (IntellectReq) that can be deployed on the edge to measure the request necessity with low resource consumption, so as to boost the efficient edge-cloud communication in DC-CDR. Technically, we design an on-edge **M**is-**R**ecommendation **D**etector (MRD) to discriminate whether the recommendation model on the edge will make wrong recommendations (mis-recommendations). When the distribution of the edge-data change, the recommendation model on the edge would generalize worse to the current data and tends to make mis-recommendations. This implies the communication revenue of updating model parameters is high due to the updated parameters can appropriately model the current

data distribution. In addition, we design a **D**istribution **M**apper (DM) that enabling the model perceive the data distribution shift possibly and determining the uncertainty in the recommendation model's understanding of the semantics of the data to further facilitate MRD module. DM consists of three parts, including the prior network, posterior network, and next item prediction network, which map different click sequences to different normal distributions rather than different features.

However, the existing recommendation datasets cannot directly train MRD model. Therefore, we reconstruct the existing four datasets as our MRD datasets without any additional annotation, which provide supervised information for MRD model training based on the pre-trained DC-CDR framework at first. After that, MRD learns the mapping relationship between the sequence used to request the model parameters last time ($s_j, j \in \{t_0, t_1, t_2, ..., t_{i-1}\}$) and the real-time sequence ($s_i, i = t_i$) to mis-recommendation label (whether prediction $\hat{y}$=label $y$).

To summarize, our contributions are four-fold:

- We propose MRD to determine whether to request parameters by detecting mis-recommendation on the edge. MRD help IntellectReq achieve high revenue under any edge-cloud communication budgets.

- We designed a Distribution Mapper to determine the uncertainty in the recommendation model's understanding of the semantics of the data to further improve IntellectReq.

- We construct four MRD datasets based on the existing recommendation dataset without any additional annotation to train IntellectReq.

- We evaluate our method with extensive experiments. Experiments demonstrate the effectiveness of our method.

## 2 RELATED WORK

**Edge-cloud Collaboration.** Edge-cloud collaboration [19] is playing an increasingly important role in deep learning. Cloud-based and on-edge machine learning are two distinct approaches with different benefits and drawbacks. Edge-cloud collaboration can take advantage of them and make them complement one another. Federated learning, such as FedAVG [9], is one of the most well-known forms of edge-cloud collaboration. Federated learning is also often used for various tasks such as multi-task learning [8, 10], etc. However, the method of federated learning for edge-cloud collaboration is relatively simple and cannot meet the needs of many practical scenarios. [18] designs multiple models with the same functions but different training processes, and a Meta Controller is used to determine which model should be used. DUET [7] draws on the idea of HyperNetwork, which can ensure that the model on the edge generalizes well to the data distribution of the current data at each moment without any training on the edge. This paper focuses on applying these parameters generation-based models to recommender systems, namely DC-CDR. DC-CDR can significantly improve the generalization ability of the edge recommendation model. However, high request frequency and low communication revenue seriously reduce the practicability.

**Sequential Recommendation.** Sequential recommendation models the user's historical behavior sequence. Previous sequential recommendation algorithm such as FPMC [11] is non-deep learning based and uses Markov decision chains to model behavioral sequences. To improve the performance of the model, recent works [3, 5, 6, 13, 15, 20] propose the sequence recommendation model based on deep learning. GRU4Rec [5] uses GRU to model behavior sequences and achieves excellent performance. DIN [20] and SASRec [6] algorithms, respectively, introduce attention and transformer into sequence recommendation, which is fast and efficient. These methods are relatively influential in both academia and industry. In practical applications, the recommendation model often needs to be deployed on the edge, which significantly restricts the number of parameters and complexity. Moreover, the environment where the recommendation model is deployed is highly real-time, which makes the edge recommendation model unable to update the model in real-time using traditional generalization methods. These restrictions reduce the generalization performance of the model and also restrict the model's performance under various data distributions. This paper studies how to reduce communication costs to yield a more efficient DC-CDR paradigm.

## 3 METHODOLOGY

We describe the proposed IntellectReq in this section by presenting each module and then introduce the learning strategy of IntellectReq.

### 3.1 Problem Formulation

In DC-CDR, we have access to a set of edges $\mathcal{D} = \{d^{(i)}\}_{i=1}^{N_d}$, where each edge with its personal i.i.d history samples $\mathcal{S}_{H^{(i)}} = \{x_{H^{(i)}}^{(j,t)} = \{u_{H^{(i)}}^{(j)}, v_{H^{(i)}}^{(j)}, s_{H^{(i)}}^{(j,t)}\}, y_{H^{(i)}}^{(j)}\}_{j=1}^{N_{H^{(i)}}}$ and real-time samples

$\mathcal{S}_{R^{(i)}} = \{x_{R^{(i)}}^{(j,t)} = \{u_{R^{(i)}}^{(j)}, v_{R^{(i)}}^{(j)}, s_{R^{(i)}}^{(j,t)}\}\}_{j=1}^{N_{R^{(i)}}}$ in the current session, where $\mathcal{N}_d$, $\mathcal{N}_{H^{(i)}}$ and $\mathcal{N}_{R^{(i)}}$ represent the number of edges, history data, and real-time data, respectively. $u$, $v$ and $s$ represent user, item and click sequence composed of items. It should be noted that $s^{(j,t)}$ represents the click sequence at moment $t$ in the $j$-th sample. The goal of DC-CDR is to generalize a trained global cloud model $\mathcal{M}_g(\cdot; \Theta_g)$ learned from $\{\mathcal{S}_{H^{(i)}}\}_{i=1}^{N_d}$ to each specific local edge model $\mathcal{M}_{d^{(i)}}(\cdot; \Theta_{d^{(i)}})$ conditioned on real-time samples $\mathcal{S}_{R^{(i)}}$, where $\Theta_g$ and $\Theta_{d^{(i)}}$ respectively denote the learned parameters for the global cloud model and local edge model.

$$\text{DC-CDR}: \underbrace{\mathcal{M}_g(\{\mathcal{S}_{H^{(i)}}\}_{i=1}^{N_d}; \Theta_g)}_{\text{Global Cloud Model}} \underset{\text{Parameters}}{\overset{\text{Data}}{\longleftrightarrow}} \underbrace{\mathcal{M}_{d^{(i)}}(\mathcal{S}_{R^{(i)}}; \Theta_{d^{(i)}})}_{\text{Local Edge Model}}. \quad (1)$$

To determine whether to request parameters from the cloud, IntellectReq uses $\mathcal{S}_{\text{MRD}}$ to learn a Mis-Recommendation Detector, which decides whether to update the edge model by the DC-CDR framework. $\mathcal{S}_{\text{MRD}}$ is the dataset constructed based on $\mathcal{S}_H$ without any additional annotations for training IntellectReq. $\Theta_{\text{MRD}}$ denotes the learned parameters for the local MRD model.

$$\text{IntellectReq}: \underbrace{\mathcal{M}_{c^{(i)}t}(\mathcal{S}_{\text{MRD}}; \Theta_{\text{MRD}})}_{\text{Local Edge Model}} \overset{\text{Control}}{\longrightarrow} (\underbrace{\mathcal{M}_g \underset{\text{Parameters}}{\overset{\text{Data}}{\longleftrightarrow}} \mathcal{M}_{d^{(i)}}}_{\text{DC-CDR}}). \quad (2)$$

Figure 3 illustrates the overview of our IntellectReq framework which consists of Mis-Recommendation Detector (MRD) and Distribution Mapper (DM) to achieve high profit under any requested budget.

### 3.2 Intelligent Parameter Request Model

We first introduce the base framework of DC-CDR, where the cloud generator model generates the dynamic parameters of the on-edge model based on real-time data. To overcome these problems, we propose Intelligent Edge-Cloud Parameter Request Model to achieve high communication revenue under any edge-cloud communication budget in DC-CDR. Specifically, we propose Mis-Recommendation Detector (MRD), which could determine whether requesting parameters from the cloud model $\mathcal{M}_g$ or using the on-edge recommendation model $\mathcal{M}_d$ based on the real-time data $\mathcal{S}_{R^{(i)}}$. And the Distribution Mapper is proposed to determine the uncertainty in the recommendation model's understanding of the semantics of the data.

*3.2.1 The framework of DC-CDR.* In this section, we will outline the edge-cloud collaboration framework DC-CDR.

In DC-CDR, a recommendation model with a backbone and a classifier will be trained for the global cloud model development. The goal of the DC-CDR can thus be formulated as the following optimization problem:

$$\hat{y}_{H^{(i)}}^{(j)} = f_{\text{rec}}(\Omega(x_{H^{(i)}}^{(j)}; \Theta_g^b); \Theta_g^c),$$
$$\mathcal{L}_{\text{rec}} = \sum_{i=1}^{N_d} \sum_{j=1}^{N_{R^{(i)}}} D_{ce}(y_{H^{(i)}}^{(j)}, \hat{y}_{H^{(i)}}^{(j)}), \quad (3)$$

where $D_{ce}(\cdot; \Theta_g^b)$ denotes the cross-entropy between two probability distributions, $f_{\text{rec}}(\cdot)$ denotes the classifier of the recommendation model, $\Omega(x_{H^{(i)}}^{(j)}; \Theta_g^b)$ is the backbone extracting features from

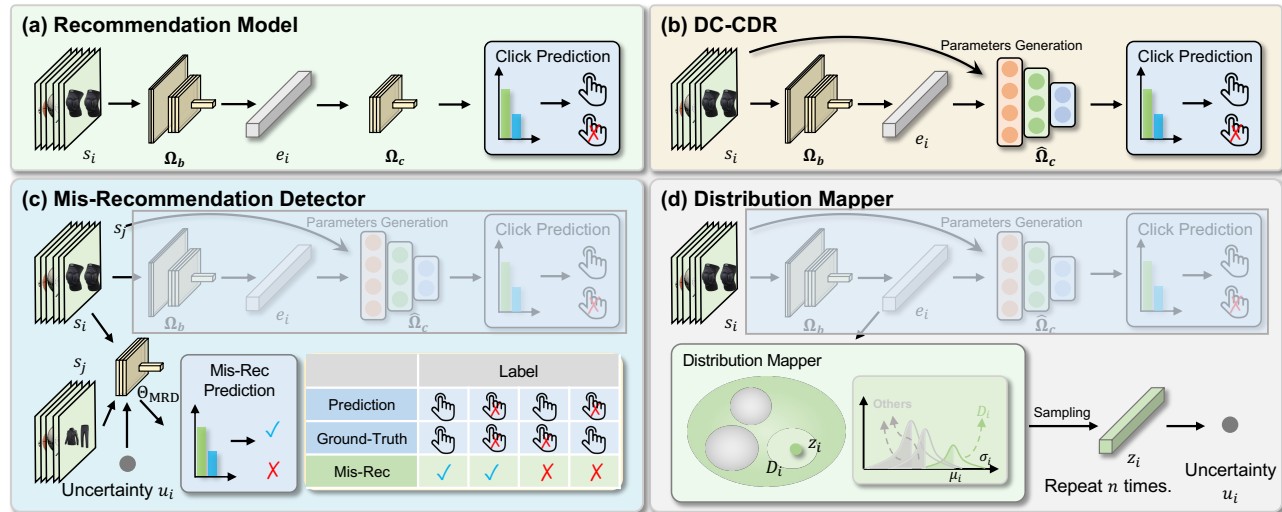

**Figure 3: Overview of the proposed IntellectReq. (a) describes the conventional recommendation model. (b) describes the DC-CDR. (c) and (d) illustrate the two modules of our IntellectReq, Mis-Recommendation Detector, and Distribution Mapper, respectively. .**

sample $x_{H^{(i)}}^{(j)}$. DC-CDR is decoupled with a backbone-classifier training scheme as modeling the "static layers" and "dynamic layers" to achieve the personalized model generalization. "Dynamic layers" is the main reason why DC-CDR can improve the generalization ability of the on-edge model to real-time data. The parameters of the backbone are fixed after finishing training as Eq. 3 and represented by $\Theta_g^b$. The parameters of the classifier are generated by the cloud generator model according to the real-time data and represented by $\Theta_g^c$.

*In edge inference*, the cloud generator model uses the real-time click sequence $s_{R^{(i)}}^{(j,t)} \in \mathcal{S}_{R^{(i)}}$ to generate the model parameters as follows,

$$h_{R^{(i)}}^{(n)} = L_{\text{layer}}^{(n)}(e_{R^{(i)}}^{(j,t)} = E_{\text{shared}}(s_{R^{(i)}}^{(j,t)})), \forall n = 1, \cdots, \mathcal{N}_l, \quad (4)$$

where $E_{\text{share}}(\cdot)$ represents the shared encoder. $L_{\text{layer}}^{(n)}(\cdot)$ is a linear layer used to adjust $e_{R^{(i)}}^{(j,t)}$ which is the output of $E_{\text{share}}(\cdot)$ to the $n^{th}$ dynamic layer features. $e_{R^{(i)}}^{(j,t)}$ means embedding vector generated by the click sequence at the moment $t$.

The cloud generator model treats the parameters of a fully-connected layer as a matrix $K^{(n)} \in \mathbb{R}^{N_{in} \times N_{out}}$, where $N_{in}$ and $N_{out}$ represent the number of input neurons and output neurons of the $n^{th}$ fully-connected layers, respectively. Then the cloud generator model $g(\cdot)$ converts the real-time click sequence $s_{R^{(i)}}^{(j,t)}$ into dynamic layers parameters $\hat{\Theta}_g^c$ by $K_{R^{(i)}}^{(n)} = g^{(n)}(e_{R^{(i)}}^{(n)})$. Since the following content no longer needs the superscript $(n)$, we simplify $g(\cdot)$ to $g(\cdot) = L_{\text{layer}}^{(n)}(E_{\text{shared}}(\cdot))$. Then, the edge recommendation model updates the parameters and makes inference as follows,

$$\hat{y}_{R^{(i)}}^{(j,t)} = f_{\text{rec}}(\Omega(x_{R^{(i)}}^{(j,t)}; \Theta_g^b); \hat{\Theta}_g^c = g(s_{R^{(i)}}^{(j,t)}; \Theta_p)). \quad (5)$$

*In cloud training*, all layers of the cloud generator model are optimized together with the static layers of the primary model that are conditioned on the global history data $\mathcal{S}_{H^{(i)}} = \{x_{H^{(i)}}^{(j)}, y_{H^{(i)}}^{(j)}\}_{j=1}^{\mathcal{N}_{H^{(i)}}}$, instead of optimizing the static layers of the primary model first and then optimizing the cloud generator model. The cloud generator model loss function is defined as follows:

$$\mathcal{L} = \sum_{i=1}^{\mathcal{N}_d} \sum_{j=1}^{\mathcal{N}_{H^{(i)}}} D_{ce}(y_{H^{(i)}}^{(j)}, \hat{y}_{H^{(i)}}^{(j)}). \quad (6)$$

DC-CDR could improve the generalization ability of the edge recommendation model. However, DC-CDR could not be easily deployed in a real-world environment due to the high request frequency and low communication revenue. Under the DC-CDR framework, the moment $t$ in Eq. 5 is equal to the current moment $T$, which means that the edge and the cloud communicate at every moment. In fact, however, a lot of communication is unnecessary because $\hat{\Theta}_g^c$ generated by the sequence earlier may work well enough. To alleviate this issue, we propose Mis-Recommendation Detector (MRD) and Distribution Mapper (DM) to solve the problem when the edge recommendation model should update parameters.

*3.2.2 Mis-Recommendation Detector.* The training procedure of MRD can be divided into two stages. The goal of the first stage is to construct a MRD dataset $\mathcal{S}_C$ based on the user's historical data without any additional annotation to train the MRD. The cloud model $\mathcal{M}_g$ and the edge model $\mathcal{M}_d$ are trained in the same way as the training procedure of DC-CDR.

$$\hat{y}_{H^{(i)}}^{(j,t,t')} = f_{\text{rec}}(\Omega(x_{H^{(i)}}^{(j,t)}; \Theta_g^b); \hat{\Theta}_g^c = g(s_{H^{(i)}}^{(j,t')}; \Theta_p)). \quad (7)$$

Here, we set $t' \le t = T$. That is, when generating model parameters, we use the click sequence $s_{R^{(i)}}^{(j,t')}$ at the previous moment $t'$, but this model is used to predict the current data. Then we can get $c^{(j,t,t')}$ that means whether the sample be correctly predicted based on the

prediction $\hat{y}_{R^{(i)}}^{(j,t,t')}$ and the ground-truth $y_{R^{(i)}}^{(j,t)}$.

$$
c^{(j,t,t')} = \left\{ \begin{array}{ll} 1, \hat{y}_{R^{(i)}}^{(j,t,t')} = y_{R^{(i)}}^{(j,t)}; \\ 0, \hat{y}_{R^{(i)}}^{(j,t,t')} \neq y_{R^{(i)}}^{(j,t)}. \end{array} \right. , \tag{8}
$$

$$
\mathcal{L}_{\text{MRD}} = \sum_{j=1}^{|\mathcal{S}_{\text{MRD}}^{(i)}|} \sum_{t'=1}^{T} l(y_j, \hat{y} = f_{\text{MRD}}(s^{(j,t)}, s^{(j,t')}; \Theta_{\text{MRD}})). \tag{9}
$$

Then we construct the new mis-recommendation training dataset as follows: $\mathcal{S}_{\text{MRD}}^{(i)} = \{s^{(j,t)}, s^{(j,t')}, c^{(j,t,t')}\}_{0 \le t' \le t=T}$. Then, a classifier $f_{\text{MRD}}(\cdot)$ can be trained on $\mathcal{S}_{\text{MRD}}^{(i)}$ according to the Eq. 9, where $t = T$ and the loss function $l(\cdot)$ is cross entropy.

### 3.2.3 Distribution Mapper.
Although the MRD could determine when to update edge parameters, it is insufficient to simply map a click sequence to a certain representation in a high-dimensional space due to ubiquitous noises in click sequences. So we design the **D**istribution **M**apper (DM) make it possible to directly perceive the data distribution shift and determine the uncertainty in the recommendation model's understanding of the semantics of the data. **The detailed architecture figure can be referred to Appendix.**

Inspired by Conditional-VAE, we map click sequences to normal distributions. Different from the original MRD, the DM module consider a variable $u^{(j,t)}$ to denote the uncertainty in Equation 9 as:

$$
\mathcal{L}_{\text{MRD}} = \sum_{j=1}^{|\mathcal{S}_{\text{MRD}}^{(i)}|} \sum_{t'=1}^{T} l(y_j, \hat{y} = f_{\text{MRD}}(s^{(j,t)}, s^{(j,t')}, u^{(j,t)}; \Theta_{\text{MRD}})). \tag{10}
$$

The uncertainty variable $u^{(j,t)}$ shows the recommendation model's understanding of the semantics of the data. DM focuses on how to learn such uncertainty variable $u^{(j,t)}$.

Distribution Mapper consists of three components as shown in the figure in Appendix, namely the **P**rior **N**etwork $P(\cdot)$ (PRN), the **P**osterior **N**etwork $Q(\cdot)$ (PON), and the **N**ext-item **P**rediction **N**etwork $f(\cdot)$ (NPN) that includes the backbone $\Omega(\cdot)$ and classifier $f_{\text{NPN}}(\cdot)$. Note that $\Omega(\cdot)$ here is the same as $\Omega(\cdot)$ in section 3.2.1 and 3.2.2, so there is almost no additional resource consumption. We will first introduce the three components separately, and then introduce the training procedure and inference procedure.

*Prior Network.* The Prior Network with weights $\Theta_{\text{prior}}$ and $\Theta'_{\text{prior}}$ maps the representation of a click sequence $s^{(j,t)}$ to a prior probability distribution. We set this prior probability distribution as a normal distribution with mean $\mu_{\text{prior}}^{(j,t)} = \Omega_{\text{prior}}(s^{(j,t)}; \Theta_{\text{prior}}) \in \mathbb{R}^N$ and variance $\sigma_{\text{prior}}^{(j,t)} = \Omega'_{\text{prior}}(s^{(j,t)}; \Theta'_{\text{prior}}) \in \mathbb{R}^N$.

$$
\mathbf{z}^{(j,t)} \sim P(\cdot|s^{(j,t)}) = \mathcal{N}(\mu_{\text{prior}}^{(j,t)}, \sigma_{\text{prior}}^{(j,t)}). \tag{11}
$$

*Posterior Network.* The Posterior Network $\Omega_{\text{post}}$ with weights $\Theta_{\text{post}}$ and $\Theta'_{\text{post}}$ can assist the training of the Prior Network by introducing posterior information. It maps the representation concatenated by the representation of the next-item $r_{(j,t)}$ and of the click sequence $s^{(j,t)}$ to a normal distribution. we set this posterior probability distribution as a normal distribution with mean $\mu_{\text{post}}^{(j,t)} = \Omega_{\text{post}}(s^{(j,t)}; \Theta_{\text{post}}) \in \mathbb{R}^N$ and variance $\sigma_{\text{post}}^{(j,t)} = \Omega'_{\text{post}}(s^{(j,t)}; \Theta'_{\text{post}}) \in \mathbb{R}^N$.

$$
\mathbf{z}^{(j,t)} \sim Q(\cdot|s^{(j,t)}, r^{(j,t)}) = \mathcal{N}(\mu_{\text{post}}^{(j,t)}, \sigma_{\text{post}}^{(j,t)}). \tag{12}
$$

*Next-item Prediction Network.* The Next-item Prediction Network with weights $\Theta_c$ predicts the embedding of the next item $\hat{r}_{(j,t)}$ to be clicked based on the user's click sequence $s^{(j,t)}$ as follows,

$$
\begin{aligned}
\hat{r}_{(j,t)} &= f_c(e^{(j,t)} = \Omega(s^{(j,t)}; \Theta_b), z^{(j,t)}; \Theta_c), \\
\hat{y}^{(j,t)} &= f_{\text{rec}}(\Omega(x^{(j,t)}; \Theta_g^b), \hat{r}^{(j,t)}; g(e^{(j,t)}; \Theta_p)).
\end{aligned} \tag{13}
$$

**Training Procedure.** In the training procedure, two losses need to be constructed, one is recommendation prediction loss $\mathcal{L}_{rec}$ and the other is distribution difference loss $\mathcal{L}_{dist}$. Like the way that most recommendation models are trained, $\mathcal{L}_{rec}$ uses the binary cross-entropy loss function $l(\cdot)$ to penalize the difference between $\hat{y}^{(j,t)}$ and $y^{(j,t)}$. The difference is that here NPN uses the feature $z$ sampled from the prior distribution $Q$ to replace $e$ in formula 5 In addition, $\mathcal{L}_{dist}$ penalizes the difference between the posterior distribution $Q$ and the prior distribution $P$ with the help of the Kullback-Leibler divergence. $\mathcal{L}_{dist}$ "pulls" the posterior and prior distributions towards each other. The formulas for $\mathcal{L}_{rec}$ and $\mathcal{L}_{dist}$ are as follows,

$$
\mathcal{L}_{rec} = \mathbb{E}_{\mathbf{z} \sim Q(\cdot|s^{(j,t)}, y^{(j,t)})}[l(y^{(j,t)}|\hat{y}^{(j,t)})], \tag{14}
$$

$$
\mathcal{L}_{dist} = D_{KL}(Q(z|s^{(j,t)}, y^{(j,t)})||P(z|s^{(j,t)})). \tag{15}
$$

Finally, we optimize the whole DM according to the following formula,

$$
\mathcal{L}(y^{(j,t)}, s^{(j,t)}) = \mathcal{L}_{rec} + \beta \cdot \mathcal{L}_{dist}. \tag{16}
$$

The training procedure is done from scratch using randomly initialized weights.

**Inference Procedure.** In the inference procedure, the posterior network will be removed from DM because there is no posterior information during the inference procedure. Uncertainty variable $u^{(j,t)}$ is calculated by the multi-sampling outputs as follows:

$$
u^{(j,t)} = \text{var}(\hat{r}_i = f_c(\Omega(s^{(j,t)}; \Theta_b), z_{1 \sim n}^{(j,t)}; \Theta_c)), \tag{17}
$$

where $n$ denotes the sampling times. Specifically, we consider the dimension of $\hat{r}^{(j,t)}$ is $N \times 1$, $\hat{r}_i^{(j,t),(k)}$ as the $k$-th value of the $\hat{r}_i^{(j,t)}$ vector, and calculate the variance as follows:

$$
\text{var}(\hat{r}_i) = \sum_{k=1}^{N} \text{var}\hat{r}_{1 \sim n}^{(j,t),(k)}. \tag{18}
$$

### 3.2.4 On-device Model Update.
**M**is-**R**ecommendation **S**core (MRS) is a variable calculated based on the output of MRD and DM, which directly affects whether the model needs to be updated.

$$
\text{MRS} = 1 - f_{\text{MRD}}(s^{(j,t)}, s^{(j,t')}; \Theta_{\text{MRD}}) \tag{19}
$$

$$
\text{Update} = \mathbb{1}(\text{MRS} \le \text{Threshold}) \tag{20}
$$

In the equation above, $\mathbb{1}(\cdot)$ is the indicator function. To get the threshold, we need to collect user data for a period of time, then get the MRS values corresponding to these data on the cloud and sort them, and then set the threshold according to the load of the cloud server. For example, if the load of the cloud server needs to be reduced by 90%, that is, when the load is only 10% of the previous value, only the minimum 10% position value needs to be sent to each device as the threshold. During inference, each device determines whether it needs to update the device model based on equation 19 and 20, that is, whether it needs to request new parameters.

# 4 EXPERIMENTS

We conducted extensive experiments on three public recommendation datasets to demonstrate the effectiveness and generalizability of the proposed Intelligent Edge-Cloud Parameter Request Model. Due to space limitations, we put part of the experimental setup, results and analysis in the Appendix.

## 4.1 Experimental Setup.

**Datasets.** We evaluate IntellectReq and baselines on Amazon CDs (CDs), Amazon Electronic (Electronic), Douban Book (Book), three widely used public benchmarks in the recommendation tasks. The details of these three datasets and preprocessing methods can be found in the Appendix.

**Evaluation Metrics** In the experiments, we use the widely adopted AUC [3], UAUC [3], HitRate and NDCG as the metrics to evaluate model performance. They are defined by the following equations.

$$\text{AUC} = \frac{\sum_{x_0 \in \mathcal{D}_T} \sum_{x_1 \in \mathcal{D}_F} \mathbb{1}[f(x_1) < f(x_0)]}{|\mathcal{D}_T||\mathcal{D}_F|}, \quad (21)$$

$$\text{UAUC} = \frac{1}{|\mathcal{U}|} \sum_{u \in \mathcal{U}} \frac{\sum_{x_0 \in \mathcal{D}_T^u} \sum_{x_1 \in \mathcal{D}_F^u} \mathbb{1}[f(x_1) < f(x_0)]}{|\mathcal{D}_T^u||\mathcal{D}_F^u|}, \quad (22)$$

$$\text{NDCG@K} = \sum_{u \in \mathcal{U}} \frac{1}{|\mathcal{U}|} \frac{2^{\mathbb{1}(R_{u,g_u} \le K)} - 1}{\log_2(\mathbb{1}(R_{u,g_u} \le K) + 1)}, \quad (23)$$

$$\text{HitRate@K} = \frac{1}{|\mathcal{U}|} \sum_{u \in \mathcal{U}} \mathbb{1}(R_{u,g_u} \le K), \quad (24)$$

In the equation above, $\mathbb{1}(\cdot)$ is the indicator function. $f$ is the model to be evaluated. $R_{u,g_u}$ is the rank predicted by the model for the ground truth item $g_u$ and user $u$. $\mathcal{D}_T, \mathcal{D}_F$ is the positive and negative testing sample set, respectively, and $\mathcal{D}_T^u, \mathcal{D}_F^u$ is the positive and negative testing sample set for user $u$ respectively.

**Baselines.** To verify the applicability, the following representative sequential modeling approaches are implemented and compared with the counterparts combined with the proposed method.

**DUET** [7] and **APG** [16] are SOTA of DC-CDR, which generate parameters through the edge-cloud collaboration for different tasks. With the cloud generator model, the on-edge model could generalize well to the current data distribution in each session without training on the edge.

**GRU4Rec** [5], **DIN** [20], and **SASRec** [6] are three of the most widely used sequential recommendation methods in the academia and industry, which respectively introduce GRU, Attention, and Self-Attention into the recommendation system.

**LOF** [1] and **OC-SVM** [14] estimate the density of a given point via the ratio of the local reachability of its neighbors and itself. They can be used to detect changes in the distribution of click sequences. For the IntellectReq framework, we consider SASRec as our backbone unless otherwise stated, but note that IntellectReq broadly applies to all sequential recommendation backbones such as DIN, GRU4Rec, etc.

**Evaluation Metrics.** In the experiments, we use the widely adopted AUC, HitRate, and NDCG as the metrics to evaluate model performance. The detailed definitions of these metrics can be referred to in the Appendix.

## 4.2 Experimental Results.

---

[3]Note 0.1% absolute AUC gain is regarded as significant for the CTR task [6, 7, 16, 20]

### 4.2.1 Quantitative Results.
Figure 4 and 5 summarize the quantitative results of our framework and other methods on CDs and Electronic datasets. The experiments are based on state-of-the-art DC-CDR frameworks such as DUET and APG. As shown in Figure 4-5, we combine the parameter generation framework with three sequential recommendation models, DIN, GRU4Rec, SASRec. We evaluate these methods with AUC and UAUC metrics on CDs and Book datasets. We have the following findings: (1) The DUET framework (DUET) and the APG framework (APG) can be viewed as the upper bound of performance for all methods since DUET and APG are evaluated with fixed 100 request frequency and other methods are evaluated with increasing frequency. Note that directly comparing the other methods with DUET and APG is not fair as DUET and APG use the fixed 100 request frequency, which could not be deployed in lower request frequency. (2) The random request method (DUET (Random), APG (Random)) works well with any request budget. However, it does not give the optimal request scheme for any request budget in most cases (such as Row.1). The correlation between its performance and Request Frequency tends to be linear. The performances of random request methods are unstable and unpredictable, where these methods outperform other methods in a few cases. (3) LOF (DUET (LOF), APG (LOF)) and OC-SVM (DUET (OC-SVM), APG (OC-SVM)) are two methods that could be used as simple baselines to make the optimal request scheme under a special and specific request budget. However, they have two weaknesses. One is that they consume a lot of resources and thus significantly reduce the calculation speed. The other is they can only work under a specific request budget instead of an arbitrary request budget. For example, in the first line, the Request Frequency of OC-SVM can only be 60%. (4) In most cases, our IntellectReq can make the optimal request scheme under any request budget.

### 4.2.2 Mis-recommendation score and profit.
Figure 6 shows that the relationship between request frequency and different threshold. To further study the effectiveness of MDR, we visualize the request timing and revenue in Figure 7. As shown in Figure 7, we analyze the relationship between request and revenue. Every 100 users were assigned to one of 15 groups, which were selected at random. The Figure is divided into three parts, with the first part used to assess the request and the second and third parts used to assess the benefit. The metric used here is Mis-Recommendation Score (MRS) to evaluate the request revenue. MRS is a metric to measure whether a recommendation will be made in error. In other words, it can be viewed as an evaluation of the model's generalization ability. The probabilities of a mis-recommendation and requesting model parameters are higher and the score is lower.

- **IntellectReq**. The IntellectReq predicts the MRS based on the uncertainty and the click sequences at the moment $t$ and $t-1$.
- **DUET (Random)**. Due to DUET (Random) request to the cloud model randomly under the DUET framework, MRS can be regarded as an arbitrary constant. Here we take the mean value of the MRS of IntellectReq as the MRS value of DUET (Random).
- **DUET (w. Request)** represents the performance curve if the edges send real-time data to the cloud at the moment $t$ and update the model parameters on the edge.

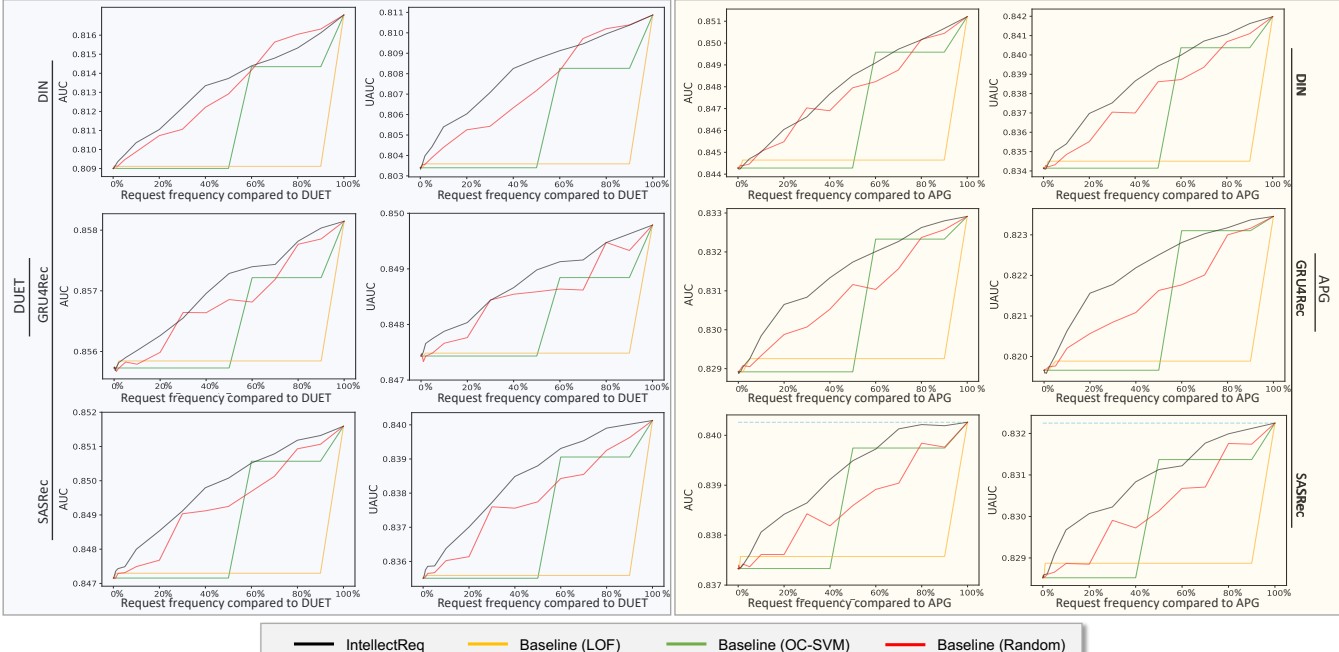

Figure 4: Performance *w.r.t.* Request Frequency curve on Amazon-CDs Dataset.

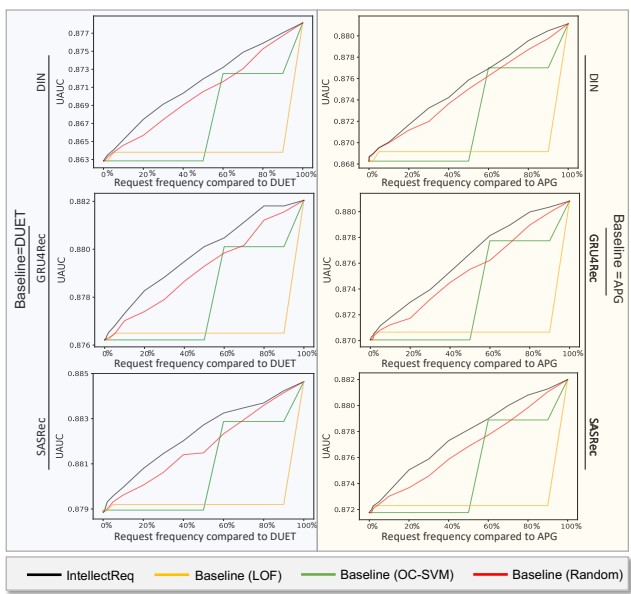

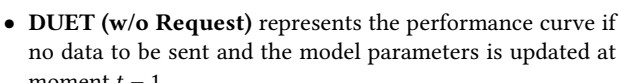
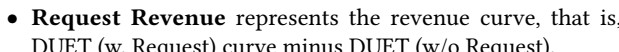

Figure 5: Performance *w.r.t.* Request Frequency curve on Douban-Book Dataset.

- **DUET (w/o Request)** represents the performance curve if no data to be sent and the model parameters is updated at moment $t - 1$.
- **Request Revenue** represents the revenue curve, that is, DUET (w. Request) curve minus DUET (w/o Request).

From Figure 7, we have the following observations: (1) The trends of MRS and DUET Revenue are typically in the opposite direction, which means that when the MRS value is low, IntellectReq tends to

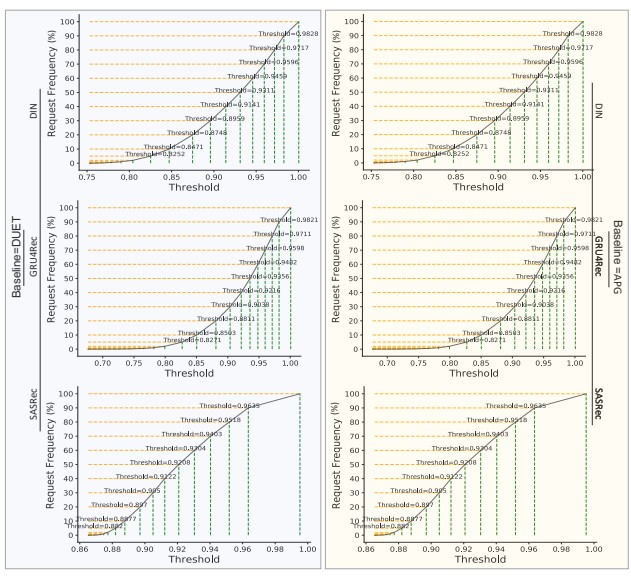

Figure 6: Request frequency *w.r.t.* different threshold

believe that the edge's model cannot generalize well to the current data distribution. Then, the IntellectReq uses the most recent real-time data to request model parameters. As a result, the revenue at this time is frequently positive and relatively high. When the MRS value is high, IntellectReq tends to continue using the model that was updated at the previous moment $t - 1$ instead of $t$ because it believes that the model on the edge can generalize well to the current data distribution. The revenue is frequently low and negative

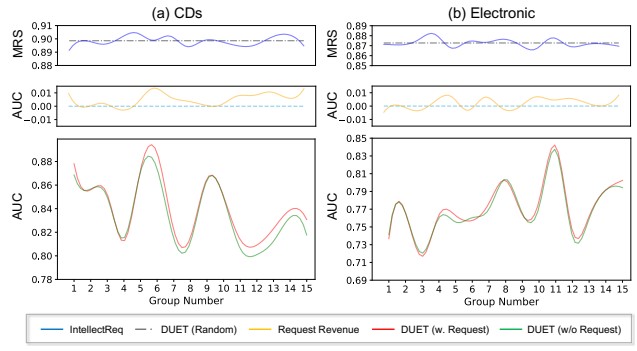

**Figure 7: Mis-Recommendation Score and Revenue.**

if the model parameters are requested at this point. (2) Since the MRS of DUET (Random) is constant, it cannot predict the revenue of each request. The performance curve changes randomly because of the irregular arrangement order of groups.

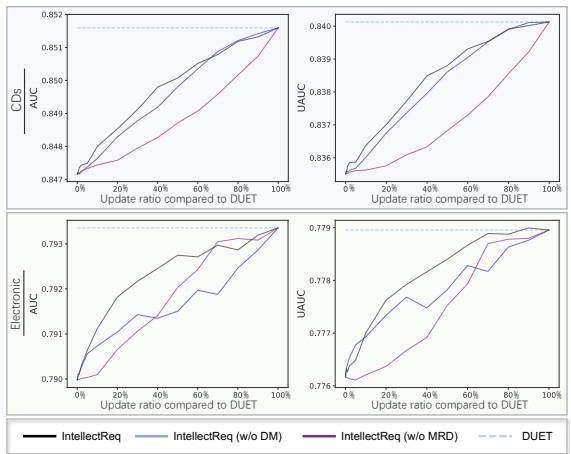

**Figure 8: Ablation study on model architecture.**

*4.2.3 Ablation Study.* We conducted an ablation study to show the effectiveness of different components in IntellectReq. The results are shown in Figure 8 .

We use w/o and w. to denote without and with, respectively. From the Table, we have the following findings:

- **IntellectReq** means both DM and MRD are used.
- **(w/o DM)** means MRD is used but DM is not used.
- **(w/o MRD)** means DM is used but MRD is not used.

From the Figure and Table, we have the following observations: (1) Generally, IntellectReq achieves the best performance with different evaluation metrics in most cases, demonstrating the effectiveness of IntellectReq. (2) When the request frequency is small, the difference between IntellectReq and IntellectReq (w/o DM) is not immediately apparent, as shown in Fig. 8(d). The difference becomes more noticeable when the Request Frequency increases within a certain range. In brief, the difference exhibits the traits of first getting smaller, then larger, and finally smaller.

*4.2.4 Time and Space Cost.* Most edges have limited storage space, so the on-edge model must be small and sufficient. The edge's computing power is rather limited, and the completion of the recommendation task on the edge requires lots of real-time processing, so the model deployed on the edge must be both simple and fast. Therefore, we analyze whether these methods are controllable and highly profitable based on the DUET framework, and additional time and space resource consumption under this framework is shown in Table 1. In the time consumption column, signal "/" separates the

**Table 1: Extra Time and Space Cost on CDs dataset.**

| Method | Controllable | Profitable | Time Cost | Space Cost (Param.) |
|---|---|---|---|---|
| LOF | ✗ | ✓ | 225s/11.3ms | $\approx 0$ |
| OC-SVM | ✗ | ✓ | 160s/9.7ms | $\approx 0$ |
| Random | ✓ | ✗ | 0s/0.8ms | $\approx 0$ |
| IntellectReq | ✓ | ✓ | 11s/7.9ms | $\approx 5.06k$ |

time consumption of cloud preprocessing and edge inference. Cloud preprocessing means that the cloud server first calculates the MRS value based on recent user data and then determines the threshold based on the communication budget of the cloud server and sends it to the edge. Edge inference refers to the MRS calculated when the click sequence on the edge is updated. The experimental results show that: 1) In terms of time consumption, both cloud preprocessing and edge inference are the fastest for random requests, followed by our IntellectReq. LOF and OC-SVM are the slowest. 2) In terms of space consumption, random, LOF, and OC-SVM can all be regarded as requiring no additional space consumption. In contrast, our method requires the additional deployment of 5.06k parameters on the edge. 3) Random and our IntellectReq can be realized in terms of controllability. It means that edge-cloud communication can be realized under the condition of an arbitrary communication budget, while LOF and OC-SVM cannot. 4) In terms of high yield, LOF, OC-SVM, and our IntellectReq can all be achieved, but random requests cannot. In general, our IntellectReq only requires minimal time consumption (does not affect real-time performance) and space consumption (easy to deploy for smart edges) and can take into account controllability and high profitability.

## 5 CONCLUSION

In this paper, we argued that most of the communications under the DC-CDR framework are unnecessary to request the new parameters of the recommendation system on the cloud since the on-edge data distribution not always changing. We designed an IntellectReq that can be deployed on the edge to calculate the request revenue with low resource consumption to alleviate this issue and ensure adaptive edge-cloud communication with high revenue. We introduce a new edge intelligence learning task to implement IntellectReq by detecting whether the data is out-of-domain. Moreover, we map the user's real-time behavior to the normal distribution and then calculate the uncertainty by the multi-sampling outputs to measure the generalization ability of the edge model to the current user behavior. Extensive experiments demonstrates IntellectReq's effectiveness and generalizability on four public benchmarks, yielding a more efficient edge-cloud collaborative and dynamic recommendation paradigm.

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

# A  APPENDIX

This is the Appendix for "Intelligent Model Update Strategy for Sequential Recommendation".

## A.1  Overview of the proposed Distribution Mapper

The overview of the proposed Distribution Mapper is shown in Figure 9

## A.2  Supplementary Experimental Results

*A.2.1  Datasets.* We evaluate IntellectReq and baselines on Amazon CDs (CDs) [4], Amazon Electronic (Electronic) [4], Douban Book (Book) [5], three widely used public benchmarks in the recommendation tasks. Following conventional practice, all user-item pairs in the dataset are treated as positive samples. In order to conduct sequential recommendation experiments, we arrange the items clicked by the user into a sequence in the order of timestamps. We also refer to [5, 6, 20], which is negatively sampled at 1 : 4 and 1 : 99 in the training set and test set, respectively. Negative sampling considers all user-item pairs that do not exist in the dataset as negative samples. The statistics of three datasets is shown in 2.

**Table 2: Statistics of Datasets.**

|  | Amazon CDs | Amazon Electronic | Douban Books |
|---|---|---|---|
| #User | 1,578,597 | 4,201,696 | 46,549 |
| #Item | 486,360 | 476,002 | 212,996 |
| #Interaction | 3,749,004 | 7,824,482 | 1,861,533 |
| #Density | 0.0000049 | 0.0000039 | 0.0002746 |

We further summarize the basic information of the MRD datasets in Table 3, which shows the accuracy and robustness.

**Table 3: Statistics of Mis-Recommendation Datasets.**

|  | Amazon CDs | Amazon Electronic | Douban Books |
|---|---|---|---|
| Accuracy | 0.9332±0.0068 | 0.9321±0.0192 | 0.9443±0.0009 |
| TPR@FPR=10e-5 | 0.9998±0.0002 | 0.9998±0.0002 | 0.9999±0.0000 |
| TPR@FPR=10e-6 | 1.0000±0.0001 | 1.0000±0.0000 | 1.0000±0.0000 |

## A.3  Training Procedure and Inference Procedure

In this section, we describe the overall pipeline in detail in conjunction with Figure 10.

1. **Training Procedure**

① We first pre-trained a DC-CDR framework, and DC-CDR can use data to generate model parameters.

② **MRD training procedure**. 1) **Construct the MRD dataset**. We assume that the time at this time is $T$, and then we use the model parameters generated by the data at moment $t = 0$ under the DC-CDR framework, and the model is applied to the data at the current moment $t = T$. At this point, we can get a prediction result $\hat{y}$,

---

[4]https://jmcauley.ucsd.edu/data/amazon/
[5]https://www.kaggle.com/datasets/fengzhujoey/douban-datasetratingreviewside-information

compare $\hat{y}$ with $y$ to get whether the model do mis-recommendation. We then repeat the data used for parameter generation from $t = 0$ to $t = T - 1$, which constructs an MRD dataset. It contains three columns, namely: the data used for parameter generation ($x_1$), the current data ($x_2$), and whether it do mis-recommendation ($y_{\mathrm{MRD}}$). 2) **Train MRD**. MRD is a fully connected neural network that takes $x_1$ and $x_2$ as input and fits the mis-recommendation label $y_{\mathrm{MRD}}$. And then we get the MRD. MRD can be used to determine whether the model parameters generated using the data at a certain moment before are still valid for the current data. The prediction result output by MRD can be simply considered as Mis-Recommendation Score (MRS).

③ **DM training procedure**. We map the data into a Gaussian distribution through the Conditional-VAE method, and then sample the feature vector from the distribution to complete the next-item prediction task, that is, to predict the item that the user will click next. Then we can get DM. DM can calculate multiple next-items by sampling from the distribution multiple times, which can be used to calculate Uncertainty.

④ **Joint training procedure of MRD and DM**. We use a fully connected neural network, denoted as $f(\cdot)$, and use MRS and Uncertainty as input to fit $y_{\mathrm{MRD}}$ in the MRD dataset, which is the Mis-Recommendation Label.

2. **Inference Procedure**

The MRS is calculated using all recent user data on the cloud, and the threshold of the MRS is determined according to the load. Then send this threshold to each edge. The edge has updated the model at a certain moment $t = n, n < T$ before, and now whether it is necessary to continue to update the model at moment $t = T$, that is, whether the model is invalid for the current data distribution? We only need to input the MRD and Uncertainty calculated by the data at the moment $t = n$ data and the data at the moment $t = T$ into $f(\cdot)$ for determine. In fact, what we output is a invalid degree, which is a continuous value between 0 and 1. Whether to update the edge model depends on the threshold calculated on the cloud based on the load.

## A.4  Hyperparameters and Training Schedules

We summarize the hyperparameters and training schedules of IntellectReq on the three datasets in Table 4.

**Table 4: Hyperparameters and training schedules of IntellectReq.**

| Dataset | Parameters | Setting |
|---|---|---|
| Amazon CDs Amazon Electronic Douban Book | GPU | Tesla A100 |
|  | Optimizer | Adam |
|  | Learning rate | 0.001 |
|  | Batch size | 1024 |
|  | Sequence length | 30 |
|  | the Dimension of $z$ | 1×64 |
|  | $N$ | 32 |
|  | $n$ | 10 |

Table 5 is the supplementary result of the Figure 8.

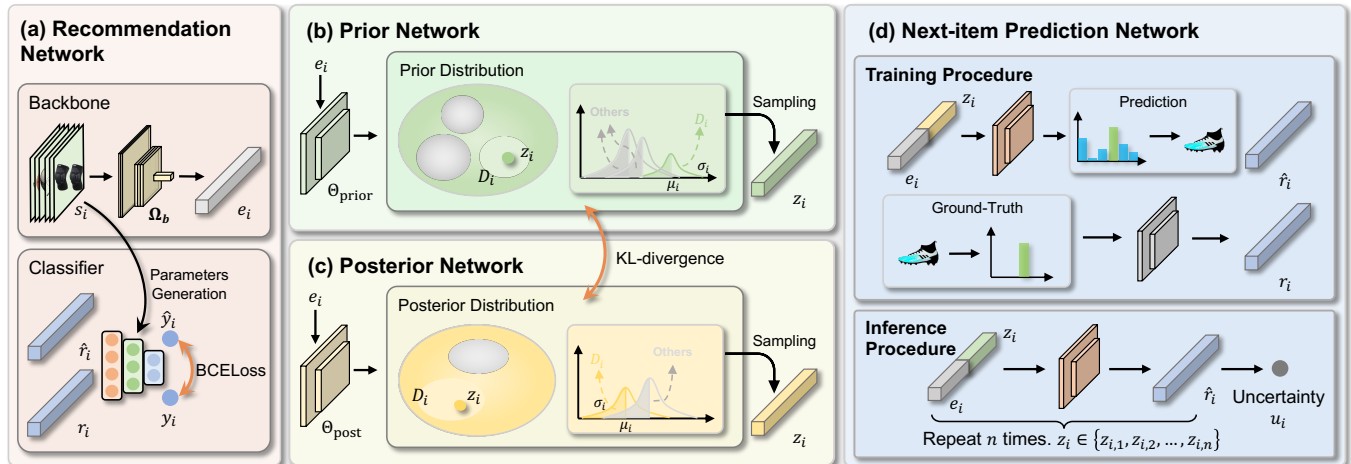

Figure 9: Overview of the proposed Distribution Mapper. Training procedure: The architecture includes Recommendation Network, Prior Network, Posterior network and Next-item Perdition Network. Loss consists of the classification loss and the KL-Divergence loss. Inference procedure: The architecture includes Recommendation Network, Prior Network and Next-item Perdition Network. The uncertainty is calculated by the multi-sampling output.

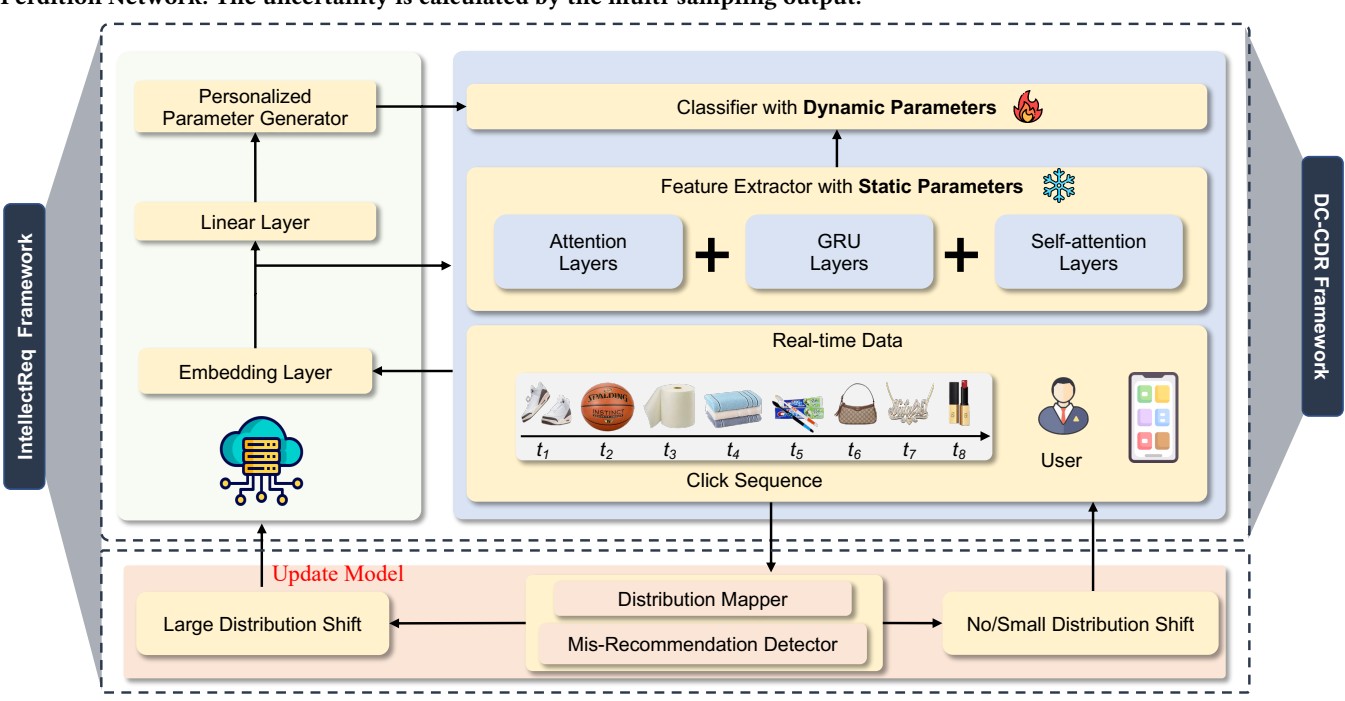

Figure 10: The overall pipeline of our proposed IntellectReq.

*A.4.1 Training Strategies and Performance.* As shown in Table. 6, we compare the impact of training DM with various loss functions on performance. When training the next-item prediction network of DM, two kinds of losses can be used: Regression Loss (RL) and Classification Loss (CL). After the DM finished training, two uncertainty calculation methods can be used, namely Mis-recommendation score Uncertainty (MU) and Next-item embedding Uncertainty (NU).

**Regression loss** directly compares the vector difference between $r$ and $\hat{r}$, we use the Mean Square Error Loss (MSELoss) as regression loss. **Classification loss** converts r and $\hat{r}$ into $y$ and $\hat{y}$ respectively with the classifier part of the recommendation model. We use the Binary Cross Entropy Loss (BCELoss) as classification loss. Note that only the training process differs between the two losses, not the inference process.

**Mis-recommendation score uncertainty** uses vectors sampled multiple times from the distribution of $t - 1$ time data mapping,

**Table 5: Ablation study on CDs dataset.**

| Method | AUC | UAUC | NDCG@10 | HR@10 | Request Frequency |
|---|---|---|---|---|---|
| IDEAL (w/o DM) | 0.8476 | 0.8360 | 0.3361 | 0.5656 | 10% |
| | 0.8488 | 0.8374 | 0.3380 | 0.5694 | 30% |
| | 0.8498 | 0.8386 | 0.3402 | 0.5721 | 50% |
| | 0.8509 | 0.8395 | 0.3421 | 0.5749 | 70% |
| | 0.8514 | 0.8401 | 0.3430 | 0.5759 | 90% |
| IDEAL (w/o MRD) | 0.8474 | 0.8356 | 0.3364 | 0.5665 | 10% |
| | 0.8479 | 0.8361 | 0.3373 | 0.5680 | 30% |
| | 0.8487 | 0.8368 | 0.3387 | 0.5694 | 50% |
| | 0.8496 | 0.8379 | 0.3406 | 0.5733 | 70% |
| | 0.8507 | 0.8392 | 0.3424 | 0.5745 | 90% |
| IDEAL | 0.8480 | 0.8364 | 0.3371 | 0.5680 | 10% |
| | 0.8491 | 0.8377 | 0.3390 | 0.5693 | 30% |
| | 0.8501 | 0.8388 | 0.3410 | 0.5735 | 50% |
| | 0.8508 | 0.8395 | 0.3424 | 0.5752 | 70% |
| | 0.8513 | 0.8400 | 0.3434 | 0.5766 | 90% |

**Table 6: Performance of training strategies on CDs dataset.**

| Method | AUC | UAUC | NDCG@10 | HR@10 | Request Frequency |
|---|---|---|---|---|---|
| CL+MU | 0.8476 | 0.8362 | 0.3371 | 0.5677 | 10% |
| | 0.8487 | 0.8372 | 0.3368 | 0.5662 | 30% |
| | 0.8493 | 0.8377 | 0.3393 | 0.5710 | 50% |
| | 0.8501 | 0.8388 | 0.3421 | 0.5745 | 70% |
| | 0.8512 | 0.8398 | 0.3423 | 0.5747 | 90% |
| RL+NU | 0.8474 | 0.8357 | 0.3374 | 0.5680 | 10% |
| | 0.8479 | 0.8361 | 0.3365 | 0.5676 | 30% |
| | 0.8491 | 0.8374 | 0.3402 | 0.5693 | 50% |
| | 0.8496 | 0.8379 | 0.3406 | 0.5710 | 70% |
| | 0.8510 | 0.8395 | 0.3423 | 0.5749 | 90% |
| CL+NU | 0.8480 | 0.8364 | 0.3371 | 0.5680 | 10% |
| | 0.8491 | 0.8377 | 0.3390 | 0.5693 | 30% |
| | 0.8501 | 0.8388 | 0.3410 | 0.5735 | 50% |
| | 0.8508 | 0.8395 | 0.3424 | 0.5752 | 70% |
| | 0.8513 | 0.8400 | 0.3434 | 0.5766 | 90% |

and vectors sampled multiple times from the distribution of $t$ time data mapping, in order to calculate multiple MRS, and further obtain $u_i$. **Next-item embedding uncertainty** uses vectors sampled multiple times from the distribution of the data map at time $t$, to calculate $u_i$ for multiple vectors.

As shown in Table 6, we compare the performance of CL+MU, RL+NU, and CL+NU. In most cases, CL+NU achieves the best performance, while CL+MU achieves the worst performance. Therefore, we use the CL+NU training strategy.

**Table 7: IDEAL's Impact on Real World.**

| | Google | | Alibaba | |
|---|---|---|---|---|
| | Bytes | FLOPs | Bytes | FLOPs |
| DC-CDR | 4.69GB | 152.46G | 152.46G | 1.68T |
| IDEAL | 3.79GB | 123.49G | 152.46G | 1.36T |
| Δ | 19.2% | | | |

*A.4.2 Impact on the Real World.* We found some more intuitive data and examples to show the challenge and IDEAL's impact on

the real world: (1) We calculate the number of bytes and FLOPs required to update a parameter. Bytes: 48.5kB, FLOPs: 1.53M. That is, updating a model on the device requires the transmission of 48.5kB data through device-cloud communication, and consumes 1.53M computing power of the cloud model. (2) According to the report, Google processes 99,000 clicks per second, so it needs to transmit 48.5kB99k=4.69GB per second, and consume 1.53M99k=152.46G of computing power in the cloud server. Alibaba processes 1,150,000 clicks per second, so it needs to transmit 48.5kB1150k=53.19GB per second, and consume 1.53M1150k=1.68T of computing power in the cloud server. These are not the peak value yet. Obviously, such a huge loan and computing power consumption make it hard to update the model for devices every moment especially at peak times. (3) Sometimes, the distributed nature of clouds today may can afford the computational volume, since it can call enough servers to support device-cloud collaboration. However, the huge resource consumption is impractical in real-scenario. Besides, according to our empirical study, our IDEAL can bring 21.4% resource saving when the performance is the same using the APG framework. Under the DUET framework, IDEAL can bring 16.6% resource saving when the performance is the same. Summing up, IDEAL can save 19% resources on average, which is very helpful for cost control and can facilitate the DC-CDR development in practice. The following table 7 is the comparison between our method IDEAL and DC-CDR in the amount of transmitted data and the computing power consumed on the cloud. (4) During the peak period, resources will be tight and cause freezes or even crashes. This is still in the case that DC-CDR has not been deployed yet, that is, the device-cloud communication only performs the most basic user data transmission. At this time, IDEAL can achieve better performance than DC-CDR under any resource limit $\epsilon$, or to achieve the performance that DC-CDR requires $\epsilon + 19\%$ of resources to achieve.

