# OpenReview forum: "Intelligent Model Update Strategy for Sequential Recommendation"
_ACM.org/TheWebConf/2024/Conference — TheWebConf24 Oral_

### Official Review · Reviewer_5XUx · 2023-11-13

**Novelty:** 5
**Technical Quality:** 4

**Review:**

The paper introduces a new model concerning edge-specific context-aware adaptivity. To investigate this problem, it introduces the Intelligent Edge-Cloud Parameter Request Model (IntellectReq). IntellectReq is designed to operate on edge, evaluating the cost-benefit landscape of parameter requests with minimal computation and communication overhead.

Some comments in the following:

- I believe that the author should carefully read the research works of Dietmar Jannach in the field of session-based and session-aware recommender systems to add to the related work.
* Empirical Analysis of Session-Based Recommendation Algorithms
M Ludewig, N Mauro, S Latifi, D Jannach
User Modeling and User-Adapted Interaction
* Session-aware Recommendation: A Surprising Quest for the State-of-the-art
S Latifi, N Mauro, D Jannach
Information Sciences 573, 291-315
- This is also connected to the choice of the baselines, since on some datasets Dietmar Jannach and his research group found out that some non-neural algorithms outperform for example GRU4rec.
- A section with the theoretical and practical implications is missing. It would be useful to understand the real impacts.
- It would be useful to check with a user study if the differences in the performance are perceived by real users.

**Questions:**

See issues above.

**Reviewer Confidence:**

4: The reviewer is certain that the evaluation is correct and very familiar with the relevant literature

**Scope:**

3: The work is somewhat relevant to the Web and to the track, and is of narrow interest to a sub-community

---

### Official Review · Reviewer_dWgN · 2023-11-26

**Novelty:** 5
**Technical Quality:** 5

**Review:**

### Evaluation of the Paper's Quality, Clarity, Originality, and Significance

#### Quality
The paper demonstrates high quality in its technical content and research methodology. It introduces an innovative model aimed at optimizing the efficiency of data exchanges between cloud and edge devices in dynamic recommendation systems. The approach taken to address the identified inefficiency and resource wastage issues is both practical and technically sound.

#### Clarity
The presentation in the paper is generally clear. The introduction effectively sets the stage for the problem and the proposed solution. However, some technical sections could benefit from more detailed explanations to enhance clarity, especially regarding the problem formulation and model architecture.

#### Originality
The paper's approach is original, particularly in its solution to the problem of inefficiency in edge-cloud data exchanges. The introduction of new components and the adaptation of existing datasets for training without additional annotation demonstrate a novel approach in the field.

#### Significance
The significance of the work lies in its potential to improve the efficiency and generalizability of recommendation systems, a crucial aspect in the context of modern e-commerce and social media platforms. The potential impact of this solution is underscored by its empirical validation.

### Pros and Cons

#### Pros
1. **Innovative Model**: The introduction of a novel model and its components to address inefficiency in recommendation systems.
2. **Practical Approach**: The adaptation of existing datasets for model training is a resource-efficient method.
3. **Empirical Validation**: The paper provides substantial experimental evidence to support the effectiveness of the proposed method.

#### Cons
1. **Technical Depth**: Some technical aspects of the paper could be elaborated further for a deeper understanding.
2. **Writing Style**: The coherence and flow of the paper's writing could be improved for better readability.
3. **Broader Impact Discussion**: A more thorough discussion on the broader impacts, including potential negative consequences and limitations of the proposed method, is missing.

### Summary
The paper presents a technically sound and novel approach to improving the efficiency of edge-cloud collaborative recommendation systems. Its originality is evident in the innovative model and practical data usage approach. The paper is strengthened by its empirical validation, though it could benefit from improvements in technical explanations, writing style, and a discussion on broader impacts.

**Questions:**

Comparison with Existing Models: How does Intellect Req compare with existing models like DIN, SASRec, and GRU4Rec in terms of performance and efficiency? Are there specific scenarios where Intellect Req significantly outperforms these models?

Practical Deployment and Industry Feedback: Has there been any practical deployment of the Intellect Req model in a real-world setting? If so, what has been the feedback from the industry, particularly concerning its efficiency and adaptability?

**Reviewer Confidence:**

3: The reviewer is confident but not certain that the evaluation is correct

**Scope:**

4: The work is relevant to the Web and to the track, and is of broad interest to the community

---

### Official Review · Reviewer_tCh3 · 2023-11-26

**Novelty:** 5
**Technical Quality:** 5

**Review:**

The draft propsoed a new framework namely IntellectReq to perform edge-cloud collaboration in the sequential recommendation task. The author proposed a MRD module to decide whether there is the need to perform parameter updata, also a DM to detect the uncertainty. The author conduct experiments on benchmark datasets. Experimental results showed that the proposed methods can achieve alomost the same recommendation accuracy with much lower communication costs. The major pros and cons are listed as follow:
Strong points:
1. The research question is interesting, i.e., edge-cloud collaboration in the recommendation scenario.
2. The experimental results demonstrate the effectiveness of the proposed methods.
Weak points:
1. The symbols in the draft should be presented more clearly, e.g., using a table to summarize the involved symbols to enhance the readbility.
2. The paper is more related to edge-computing. The communication beteeen the cloud and the edge is also needed for other scenarios. So what's the key difference between recommendation and other application scenario? What's the key challenge in the edge-cloud collaboration for recommendation. The author is suggested to highlight such points.
3. The edge-cloud collaboration is actully important for real-world recommender. However, the author conducted the experiments on the offline collected dataset. The author needs to prove that the experimental setting can effectively represent the real-world case.
4. Since the MRD module determines whether to perform parameter updatas, I'd like to see the performence of the MRD module, i.e., to which extent the MRD module can successfully make the right prediction.

**Questions:**

Please see the weak points 2 and 4.

**Reviewer Confidence:**

3: The reviewer is confident but not certain that the evaluation is correct

**Scope:**

3: The work is somewhat relevant to the Web and to the track, and is of narrow interest to a sub-community

---

### Official Review · Reviewer_JBE5 · 2023-11-26

**Novelty:** 4
**Technical Quality:** 5

**Review:**

This paper proposes a novel Intelligent Edge-Cloud Parameter Request Model (IntellectReq) to operate on edge, evaluating the cost-benefit landscape of parameter requests with minimal computation and communication overhead for Sequential Recommendation (SR). Earlier Device-Cloud Collaborative and Dynamic Recommendation system (DC-CDR) enables the personalized recommendation pattern for different edges and efficiently characterizes the real-time data distribution based on frequent edge-cloud communication, but it cannot be easily deployed in the real environment, due to the two key aspects as high request frequency and low communication revenue. To overcome these limitations, IntellectReq designs an on-edge Mis-Recommendation Detector (MRD) to determine whether to request parameters by detecting mis-recommendation on the edge and a Distribution Mapper (DM) to determine the uncertainty in the recommendation model’s understanding of the semantics of the data to further facilitate MRD module. The authors conduct the experiments on three real-world datasets, Amazon CDs (CDs), Amazon Electronic (Electronic), Douban Book (Book), comparing the proposed method with a list of SOTA baselines and backbones. However, the implementation of the method are not provided, and no section is devoted to the adopted hyperparameters for reproducibility purposes.

The paper in general is well-written. The introduction provides a good background and motivation for the problem and highlights the limitations of DC-CDR. The proposed IntellectReq method is clearly explained with the help of figures and examples.
The strength of the paper is that it proposes a novel strategy by using MRD to determine when the edge recommendation model should update parameters and DM to enable the model perceive the data distribution shift possibly and determine the uncertainty in the recommendation model’s understanding of the semantics of the data to further facilitate MRD module to reduce the request frequency and increase the communication revenue. However, there are a few limitations of this paper:
1. Although the proposed method reduces the request frequency, the performance of the model has also decreased;
2. The proposed strategy uses existing methods, such as using backbone for training in MRD, which lacks innovation;
3. The experiments are not reproducible, the code and the necessary details are missing;
4. There are some unclear expressions in the figure, such as the figure on the right in Figure 1(c), and there is no solid blue line in Figure 7, but the blue solid line is used on the label to represent IntellectReq.

**Questions:**

1. In the DM module, the method proposed in this paper utilizes VAE to map click sequences to normal distributions, but it does not seem to explain clearly why using this method can determine the uncertainty in the recommendation model’s understanding of the semantics of the data.
2. While HR and NDCG are mentioned in the Evaluation Metrics section, the comparison between the proposed method and baselines on HR and NDCG are not listed in the experiment, and only AUC and UAUC are listed?
3. Why use backbone as the model for training directly in the MRD module, but without proposing a new model that is more suitable for the scene?

**Reviewer Confidence:**

2: The reviewer is willing to defend the evaluation, but it is likely that the reviewer did not understand parts of the paper

**Scope:**

4: The work is relevant to the Web and to the track, and is of broad interest to the community

---

### Official Review · Reviewer_4att · 2023-11-28

**Novelty:** 5
**Technical Quality:** 5

**Review:**

The paper targets the computational load and communication overhead of edge-cloud collaboration for sequential recommendation. The focus was on tuning the learning based on out-of-distribution data from the edge to adapt to the recommendation scenarios specific to the edge.

Strengths:
- The research problem appears significant and interesting to me, though a concrete example from real applications should greatly help with the motivation.
- The paper presents the proposed approach in a systematic manner with adequate formulations to explain the details.
-

Weaknesses:
- The meaning of edge (or what the edge represents) in the dynamic recommendation scenario should be clearly defined. Though it is indicated in Figure 1 that each edge node incidents an end-user device, an edge node could mean different things (e.g., an edge server that is close to the user's geolocation in the network). A clear definition/explanation is necessary.
- The paper is primarily based on DC-CDR, with the addition of a control layer as indicated in Eq. (2). There are some novelty in there but not entirely substantial.
- The work is similar to federated learning tasks when applied to the recommendation regime. The authors may survey the related work on that topic and develop discussions to better justify the paper.

**Questions:**

I would like to see some evidence from practical applications/scenarios about the highly frequent requests to justify the motivation for the paper.

It might be better if the authors could present the previous work, DC-CDR, first, and then present their own work. Otherwise, it takes some effort to distinguish between the authors's new proposal and the original content from DC-CDR.

Why is "request frequency" not defined as an evaluation metric?

**Ethics Review Description:**

no concerns

**Reviewer Confidence:**

3: The reviewer is confident but not certain that the evaluation is correct

**Scope:**

3: The work is somewhat relevant to the Web and to the track, and is of narrow interest to a sub-community

---

### Decision · Program_Chairs · 2024-01-22

**Decision:**

Accept (Oral)

**Comment:**

All reviewers were generally quite positive about this paper, both in terms of novelty and technical quality. Additional explanations and more details were provided by the authors in the discussion phase. Code and data are shared by the authors for reproducibility. Overall, this work could be a solid contribution to the conference.